# Effect of intravenous vitamin C administration on chemotherapy-induced adverse events in patients with nasopharyngeal cancer

**Peesit Leelasawatsuk, Pasawat Supanimitjaroenporn, Nattida Rodsom, Theepat Wongkittithaworn, Manupol Tangthongkum**ⓘ*

Department of Otolaryngology Head and Neck Surgery, Faculty of Medicine, Prince of Songkla University, Hat Yai, Songkhla, Thailand

* tmanupol@gmail.com

## Abstract

Nasopharyngeal carcinoma is prevalent in Thailand, with a substantial proportion of cases diagnosed at advanced stages. The standard treatment, concurrent chemoradiotherapy, is associated with considerable adverse effects, which may compromise therapeutic efficacy and diminish patients' quality of life. While vitamin C has shown potential in reducing chemotherapy-induced toxicities in some cancers, its effects in nasopharyngeal carcinoma remain unclear. In this randomized, double-blind, placebo-controlled trial, patients with nasopharyngeal carcinoma undergoing concurrent chemoradiotherapy were assigned to receive either 2 g of intravenous vitamin C or placebo prior to chemotherapy. The incidence of gastrointestinal adverse effects—including nausea, anorexia, mucositis, diarrhea, and dysphagia—did not differ significantly between groups. However, longitudinal analysis demonstrated a significantly attenuated decline in platelet counts in the vitamin C group compared with placebo. Although intravenous vitamin C did not reduce gastrointestinal toxicities, the observed platelet preservation suggests a potential supportive effect that warrants further investigation.

## Trial registration

The study was registered with the Thai Clinical Trial Registry (TCTR20190316003) on March 16, 2019.

## Introduction

Nasopharyngeal carcinoma (NPC) is an epithelial malignancy arising in the nasopharynx, a unique subsite of the head and neck region. NPC is geographically distributed in a distinct manner, with the highest incidence observed in East and Southeast Asia, including Southern China, Malaysia, Vietnam, and Thailand [1]. In

**Data availability statement:** The data contain potentially identifying patient information and are subject to restrictions imposed by the Ethics Committee of the Faculty of Medicine, Prince of Songkla University, to protect participant privacy. Qualified researchers who meet the criteria for access to confidential data may request access by contacting the Research Ethics Committee, Faculty of Medicine, Prince of Songkla University (email: medpsu.ec@gmail.com).

**Funding:** This research was financially supported by a grant from the Faculty of Medicine, Prince of Songkla University, Thailand.

**Competing interests:** The authors have no conflicts of interest to declare.

these regions, NPC is a major public health concern, especially among men, with incidence rates reaching up to 10–20 cases per 100,000 individuals in endemic areas [2,3]. In Thailand, NPC accounts for approximately 1–3% of all cancers but is a leading cause of cancer-related morbidity in men aged 30–60 years [2]. NPC presents a significant clinical challenge owing to its deep anatomical location and nonspecific early symptoms—such as nasal congestion, epistaxis, headache, and hearing issues—which frequently lead to delayed diagnosis. Consequently, over 70% of cases are detected at advanced stages (stages III–IV), where prognosis is substantially worse than that in early-stage disease. While early-stage NPC has a 5-year overall survival (OS) exceeding 80%, advanced-stage disease may reduce OS to below 60% [4,5]. The primary treatment for NPC is concurrent chemoradiotherapy (CCRT), which combines radiotherapy with platinum-based chemotherapy (CMT) [6]. While CCRT has demonstrated efficacy in improving survival outcomes, it is often associated with various toxicities [7,8]. Common adverse effects include gastrointestinal symptoms, such as nausea, anorexia, oral mucositis, diarrhea, and dysphagia, hematological toxicities (e.g., neutropenia, anemia, and thrombocytopenia), and electrolyte imbalances (e.g., hyponatremia and hypokalemia) [7,9]. These adverse effects may necessitate dose reductions or lead to treatment delays and interruptions, thereby potentially compromising therapeutic effectiveness and overall survival.

Vitamin C (ascorbic acid) has garnered attention as a potential supportive therapy in oncology due to its antioxidant and anti-inflammatory properties. Since the 1970s, vitamin C has been investigated as a supportive treatment in oncology, with accumulating evidence suggesting its ability to alleviate chemotherapy-related symptoms such as nausea, vomiting, fatigue, insomnia, appetite loss, and cancer-related pain [10]. Moreover, intravenous vitamin C (IVC) has been explored for its potential role in mitigating chemotherapy-induced hematologic toxicities, particularly thrombocytopenia, through reduction of oxidative stress within the bone marrow microenvironment and preservation of megakaryocyte function [11,12]. For example, studies in ovarian cancer have reported reduced bone marrow toxicity and improved platelet recovery with adjunctive IVC administration [11], while investigations in hematologic malignancies have suggested potential restoration of hematopoietic function, including platelet counts [12]. However, findings across cancer types remain heterogeneous, and evidence in head and neck cancers, including nasopharyngeal carcinoma, is limited. The rationale for considering vitamin C as a supportive intervention in oncology is grounded in its antioxidant and immunomodulatory properties. Chemotherapy-related toxicities are, in part, driven by increased oxidative burden and activation of inflammatory pathways, which contribute to mucosal damage and suppression of hematopoiesis. By attenuating oxidative stress and regulating inflammatory signaling, IVC may help mitigate treatment-related cellular injury and support the maintenance of hematologic stability during cytotoxic therapy [13,14]. Pharmacokinetic studies have demonstrated that intravenous administration produces markedly higher plasma ascorbate concentrations than oral supplementation due to limitations in intestinal absorption and renal clearance. Padayatty et al. further clarified these differences, showing that oral administration produces only moderate plasma concentrations,

whereas intravenous infusion results in rapid and markedly elevated levels [15,16]. These pharmacokinetic properties provide a rationale for the use of IVC rather than oral supplementation in the context of chemotherapy-related toxicity management. Recent studies have largely focused on IVC, with doses ranging from 1 to 10 g per session. The frequency of administration ranges from once per chemotherapy cycle to several infusions per week, depending on study design and clinical protocol. These regimens have shown potential benefits in reducing chemotherapy-related toxicities and improving quality of life in patients with ovarian, colorectal, and hematological malignancies [10,11]. Safe IVC administration has been well documented when delivered by slow infusion under appropriate monitoring, with no harm to normal tissues, further supporting its application in oncology [11,17].

Despite growing evidence supporting the role of IVC in oncology supportive care, its specific effects in patients with NPC undergoing CCRT remain largely unexplored. Particularly, the impact of IVC on gastrointestinal toxicities during CCRT has not been well characterized in this population. Furthermore, its potential effects on hematologic parameters—including platelet preservation—and electrolyte abnormalities during cisplatin-based therapy have not been comprehensively elucidated [10]. Notably, randomized controlled trials evaluating IVC in this specific clinical setting remain scarce. Therefore, we conducted a prospective, randomized, double-blind, placebo-controlled trial to evaluate whether IVC reduces chemotherapy-induced gastrointestinal adverse events in patients with nasopharyngeal carcinoma undergoing CCRT, with secondary exploratory analyses of hematologic and electrolyte outcomes.

## Materials and methods

### Study design

This prospective, randomized, double-blind, placebo-controlled trial was conducted at Prince of Songkla University, Thailand, between March 14, 2019 and December 20, 2020. The study protocol was approved by the Ethics Committee of the Faculty of Medicine, Prince of Songkla University (REC.61-416-13-1) and registered with the Thai Clinical Trial Registry (TCTR20190316003) on March 16, 2019. The study was submitted to the Thai Clinical Trial Registry at study initiation; however, due to administrative review and clarification requests, the official registration date appears after enrollment began. Ethics approval and the finalized protocol were completed prior to participant enrollment. The authors confirm that all ongoing and related trials for this intervention are registered. The study protocol adhered to relevant guidelines and regulations, including the World Medical Association Declaration of Helsinki on ethical principles for medical research involving human participants.

### Study participants

Sixty-eight patients diagnosed with NPC and scheduled for CCRT were enrolled. The inclusion criteria required participants to be aged 18 years or older, have a confirmed NPC diagnosis, clinical stage II–IVa according to the American Joint Committee on Cancer (AJCC) Cancer Staging Manual, 8th edition, and be eligible for CCRT. The exclusion criteria included a history of hypersensitivity to vitamin C, contraindications to chemotherapy, glucose-6-phosphate dehydrogenase (G-6-PD) deficiency (confirmed through laboratory testing), a history of kidney stones, and previous radiation therapy. Participant flow was monitored throughout the study. Reasons for loss to follow-up were documented, including one withdrawal and one transferred to a different hospital in the Vitamin C group, and one treatment refusal in the placebo group. All remaining participants completed the planned course of chemoradiotherapy and intervention.

### Randomization and intervention

All participants provided written informed consent before enrollment. Randomization was performed using a computer-generated random sequence with a 1:1 allocation ratio. Allocation concealment was ensured using sequentially numbered, opaque, sealed envelopes prepared by an independent investigator not involved in patient recruitment or outcome

assessment. Group A received 2 g of IVC 1 h before chemotherapy. The IVC formulation used in this study is an approved pharmaceutical product registered with the Thai Food and Drug Administration (registration number: 1A 1557/30). The preparation complies with national regulatory standards and was administered in accordance with standard institutional safety monitoring procedures. The preparation involved removing 8 mL of 5% dextrose in water (DW) and replacing it with 2 g (8 mL) of vitamin C, resulting in a final volume of 50 mL, which was infused over 1 h. Group B received a placebo consisting of 50 mL of 5% DW without vitamin C administered over the same duration. The vitamin C and placebo solutions were identical in appearance, color, and volume. To ensure blinding, chemotherapy nurses not involved in the study prepared the solutions. The study was double-blinded, with participants, treating physicians, outcome assessors, and data analysts unaware of group allocation.

All patients underwent CCRT, consisting of platinum-based chemotherapy and external beam radiation. The chemotherapy regimen included three cycles of intravenous cisplatin (100 mg/m²) administered every three weeks. Vitamin C or placebo was administered only on the days that cisplatin was delivered. If chemotherapy was delayed, the corresponding dose of vitamin C or placebo was also withheld. All patients who completed chemotherapy received all three planned doses of the assigned intervention. Radiotherapy was delivered using intensity-modulated radiation therapy, with a total dose of 70 Gy in 35 fractions over seven weeks. The radiotherapy protocol was standardized and consistent across both treatment groups.

## Data collection

Data collected included patient demographics (sex, age, and weight), cancer histologic type, tumor differentiation, and cancer stage (following the AJCC Cancer Staging Manual, 8th edition). Information on chemotherapy dates and cycles was also recorded. Adverse effects were evaluated both prior to each chemotherapy cycle and at nadir time points (days 7–14 post-infusion), in accordance with the institutional protocol, using the Common Terminology Criteria for Adverse Events (CTCAE) version 5.0. Physical examinations and laboratory tests were conducted to monitor adverse effects. The collected parameters included gastrointestinal symptoms such as nausea, anorexia, oral mucositis, diarrhea, and dysphagia, white blood cell (WBC) counts, hemoglobin levels, platelet counts, creatinine levels, serum sodium and potassium levels. Adverse events were classified from Grade 1 (mild) to Grade 5 (death) according to CTCAE version 5.0 [18]. The primary outcome was the incidence of gastrointestinal adverse events graded according to CTCAE version 5.0 after the chemotherapy cycles. Hematologic (white blood cell count, hemoglobin level, and platelet count) and biochemical parameters (serum creatinine, sodium, and potassium levels) were classified as exploratory secondary outcomes.

## Statistical analysis

All analyses were performed according to the per-protocol approach using R software (R Foundation for Statistical Computing, Vienna, Austria). An intention-to-treat analysis was not performed due to the minimal loss to follow-up and complete outcome data among randomized participants. Descriptive statistics were summarized as frequencies, means with standard deviations (SDs), or medians with interquartile ranges (IQRs), as appropriate. Between-group comparisons for categorical variables were performed using the chi-square test or Fisher's exact test, as appropriate. Continuous variables were analyzed using independent $t$-tests or nonparametric equivalents when normality assumptions were not met. Longitudinal laboratory parameters were analyzed using linear mixed-effects models with random intercepts to account for within-subject correlations across repeated measurements. Maximum likelihood estimation was used for parameter estimation. For graphical presentation, observed median values with IQRs were displayed without baseline normalization or delta transformation. Inter-patient variability was addressed through the mixed-effects modeling framework.

The sample size was calculated to detect a reduction in the incidence of gastrointestinal adverse events. The calculation was based on the proportions reported in a previous study by Ma et al. [11], in which gastrointestinal adverse events occurred in 9 of 13 patients (69%) in the vitamin C group and in 12 of 12 patients (100%) in the control group,

corresponding to an anticipated absolute risk reduction of approximately 31%. Using these proportions, with 80% power and a two-sided alpha of 0.05, the required sample size was estimated to be 22 participants per group. To account for potential attrition, additional participants were enrolled. In addition to frequentist analyses, Bayesian logistic regression models were fitted using weakly informative priors to estimate posterior probability distributions of treatment effects. This approach allows direct probabilistic interpretation of the direction and magnitude of treatment effects, which is particularly useful in the context of exploratory secondary outcomes. Results are summarized using posterior probabilities of benefit or harm. Treatment effects were interpreted according to whether the posterior distribution indicated a high probability of benefit or harm. The Bayesian analysis was included as a complementary exploratory framework to provide additional inferential insight.

## Results

### Participant enrollment and flow

Between March 2019 and December 2020, 68 patients were enrolled in the study. Five patients were excluded owing to G-6-PD deficiency, resulting in randomization of 63 participants: 32 to the IVC group and 31 to the placebo group. Of these, 60 patients completed the full course of treatment, with 30 in each group. In the IVC group, one patient withdrew consent, and another transferred to a different hospital. In the placebo group, one patient declined to initiate CCRT. No patients discontinued the intervention because of adverse effects, and no recurrences or deaths were observed during the study period. The CONSORT diagram (Fig 1) summarizes patient enrollment, randomization, and treatment completion.

### Baseline characteristics

Data from 60 patients (19 females, 31.7% and 41 males, 68.3%) were analyzed. The mean age of the patients was 50 years and most were male. Common comorbidities included hypertension, diabetes mellitus, and dyslipidemia, with similar prevalence across both groups. Most patients were diagnosed with advanced-stage nasopharyngeal carcinoma (stage III or IVa). Baseline characteristics were similar between the two groups (Table 1).

### Gastrointestinal adverse events

The incidence of Grade 1–2 gastrointestinal adverse events did not differ significantly between the vitamin C and placebo groups. Specifically, no significant between-group differences were observed for nausea (p = 0.23), anorexia (p = 0.89), oral mucositis (p = 1.00), diarrhea (p = 0.24), or dysphagia (p = 0.24). Notably, only a small number of patients experienced Grade 3 events, and no Grade 4 toxicities were observed in our study (Table 2).

### Hematologic and biochemical parameters

In longitudinal mixed-effects analyses, significant group-by-time interaction effects were observed for platelet counts (p = 0.023) and serum sodium levels (p = 0.021), indicating differential changes over time between groups. Fig 2 displays the median and interquartile ranges (IQRs) before and after chemotherapy. The median platelet count decreased from $313 \times 10^3$ cells/μL to $276 \times 10^3$ cells/μL in the vitamin C group, compared with a decrease from $306 \times 10^3$ cells/μL to $220 \times 10^3$ cells/μL in the placebo group. Similarly, median serum sodium levels remained relatively stable in the vitamin C group (136.8 to 137.9 mmol/L), whereas a decline was observed in the placebo group (138.8 to 134.2 mmol/L).

Bayesian logistic regression demonstrated a high posterior probability of association between IVC administration and platelet count changes (P(β < 0 | data) > 0.975), whereas evidence for serum sodium changes was less consistent (posterior probability ≈ 0.50). No significant differences were observed for other hematologic or biochemical parameters.

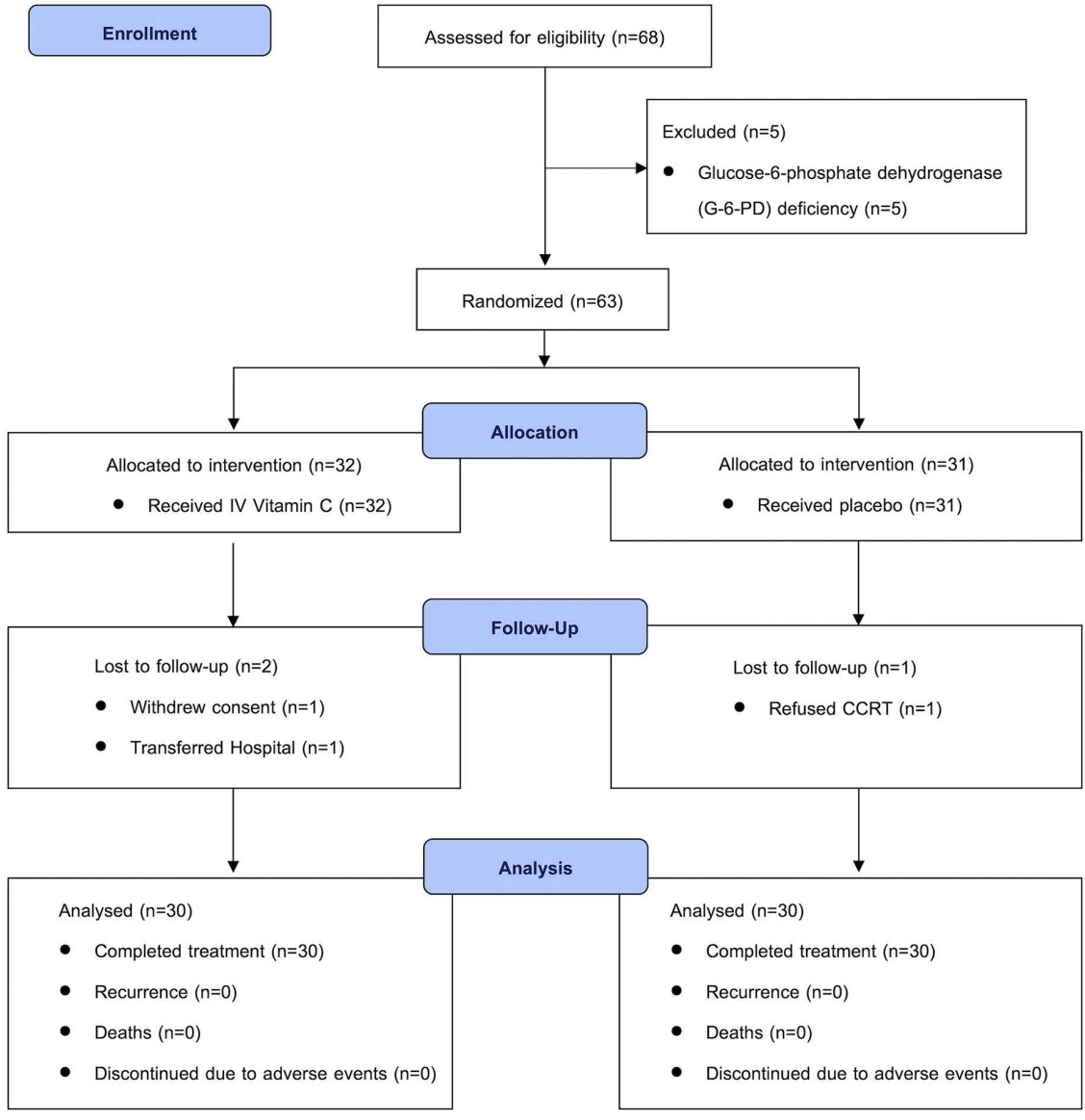

**Fig 1. CONSORT flow diagram of patient enrollment, randomization, allocation, follow-up, and analysis.**

## Discussion

Chemotherapy-induced side effects can substantially compromise treatment efficacy and diminish overall survival in patients with nasopharyngeal carcinoma. In this randomized, double-blind, placebo-controlled trial, IVC administered prior to cisplatin-based chemoradiotherapy did not significantly reduce the incidence of Grade 1–2 gastrointestinal adverse events, nor did it significantly affect other hematologic parameters (e.g., white blood cell count and hemoglobin level) or metabolic indices. However, longitudinal analysis demonstrated a significantly attenuated decline in platelet counts in the vitamin C group compared with placebo.

Previous studies investigating the effects of vitamin C on chemotherapy-related gastrointestinal toxicities have reported similar findings; the absence of significant differences in our study aligns with these earlier observations. For example,

**Table 1. Patient demographics.**

| Patient characteristics | Vitamin C (n = 30) | Placebo (n = 30) |
|---|---|---|
| **Sex, n (%)** | | |
| Female | 8 (26.7) | 11 (36.7) |
| Male | 22 (73.3) | 19 (63.3) |
| **Age, mean ± SD (range)** | 50.3 ± 12.5 (20–71) | 50.3 ± 13.0 (20–73) |
| **Underlying disease, n (%)** | | |
| Hypertension | 6 (20.0) | 4 (13.3) |
| Diabetes mellitus | 2 (6.7) | 4 (13.3) |
| Dyslipidemia | 2 (6.7) | 3 (10.0) |
| Cardiovascular disease | 1 (3.3) | 1 (3.3) |
| Pulmonary disease | 0 (0.0) | 1 (3.3) |
| Hematologic disease | 0 (0.0) | 1 (3.3) |
| HIV | 0 (0.0) | 1 (3.3) |
| Others | 1 (3.3) | 1 (3.3) |
| **Stage, n (%)** | | |
| II | 3 (10.0) | 2 (6.7) |
| III | 9 (30.0) | 8 (26.7) |
| IVa | 18 (60.0) | 20 (66.7) |
| **Pretreatment laboratory values** | | |
| WBC (cells/µL), median (IQR) (range) | 7470 (5555–9580) (3350–21280) | 7070 (6230–8350) (4080–15280) |
| Hemoglobin (g/dL), mean ± SD (range) | 12.4 ± 1.9 (8.9–16.5) | 12.6 ± 1.7 (8.2–15.6) |
| Platelet (×10³ cells/µL), median (IQR) (range) | 312 (272–366) (143–649) | 306 (256–349) (211–644) |
| Creatinine (mg/dL), mean ± SD (range) | 0.8 ± 0.3 (0.40–1.62) | 0.9 ± 0.3 (0.55–1.68) |
| Sodium (mmol/L), mean ± SD (range) | 136.9 ± 3.2 (130.1–142.0) | 138.2 ± 3.2 (128.8–143.5) |
| Potassium (mmol/L), mean ± SD (range) | 3.9 ± 0.4 (3.07–5.00) | 3.9 ± 0.4 (2.75–5.19) |

Abbreviations: HIV, human immunodeficiency virus; SD, standard deviation; IQR, interquartile range; WBC, white blood cell Continuous variables are presented as mean ± SD for approximately normally distributed data and as median (IQR) for non-normally distributed data.

Wang et al. observed no reduction in gastrointestinal adverse events in patients with colorectal cancer treated with high-dose vitamin C alongside FOLFOX-based chemotherapy [19], while Pathak et al. found no clinical benefit of high-dose antioxidants, including vitamin C, in patients with lung cancer receiving chemotherapy [20]. Similarly, Zasowska-Nowak et al. reported insufficient data to draw definitive conclusions regarding the effectiveness of IVC therapy in reducing chemotherapy-induced toxicity in patients with advanced-stage cancer [21]. The lack of observed effects in our study may be explained by several factors. First, most adverse events were of low grade (Grade 1), which may have limited the observable benefit of any supportive intervention. Second, gastrointestinal symptoms during chemoradiotherapy result from multifactorial mechanisms, including mucosal inflammation, microbiome disruption, and cytokine dysregulation, which may not be fully mitigated by the antioxidant or anti-inflammatory actions of vitamin C alone [22–24].

Despite the modest clinical differences, our study found a smaller reduction in platelet counts during CCRT in patients with NPC who received IVC than in those in the placebo group. This observation aligns with previous research findings suggesting that vitamin C may help mitigate chemotherapy-induced hematological toxicities. Foster et al. studied the effects of IVC in patients with acute myeloid leukemia undergoing chemotherapy and found improvements in hemato-logical indices, including platelet counts, suggesting that vitamin C may help restore normal hematologic function [12].

**Table 2. Highest grades of chemotherapy-induced adverse effects observed in individual patients following three cycles of chemotherapy.**

| Adverse event grade | Vitamin C group n/total (%) | Placebo group n/total (%) | p-value |
|---|---|---|---|
| Nausea | | | 0.23§ |
| No adverse event | 9/30 (30.00) | 15/30 (50.00) | |
| Grade 1 | 17/30 (56.67) | 11/30 (36.67) | |
| Grade 2 | 4/30 (13.33) | 4/30 (13.33) | |
| Anorexia | | | 0.89§ |
| No adverse event | 10/30 (33.33) | 12/30 (40.00) | |
| Grade 1 | 19/30 (63.33) | 17/30 (56.67) | |
| Grade 2 | 1/30 (3.33) | 1/30 (3.33) | |
| Oral mucositis | | | 1.00§ |
| No adverse event | 23/30 (76.67) | 23/30 (76.67) | |
| Grade 1 | 0/30 (0.00) | 0/30 (0.00) | |
| Grade 2 | 4/30 (13.33) | 4/30 (13.33) | |
| Grade 3 | 3/30 (10.00) | 3/30 (10.00) | |
| Diarrhea | | | 0.24§ |
| No adverse event | 27/30 (90.00) | 30/30 (100.00) | |
| Grade 1 | 1/30 (3.33) | 0/30 (0.00) | |
| Grade 2 | 2/30 (6.67) | 0/30 (0.00) | |
| Dysphagia | | | 0.24§ |
| No adverse event | 27/30 (90.00) | 30/30 (100.00) | |
| Grade 1 | 3/30 (10.00) | 0/30 (0.00) | |
| Grade 2 | 0/30 (0.00) | 0/30 (0.00) | |

§ Fisher's exact test

Similarly, Monti et al. conducted a clinical trial involving patients with pancreatic cancer undergoing chemotherapy and found that most patients who received IVC experienced only Grade 1 thrombocytopenia. They proposed that ascorbate may not disturb rapidly dividing normal cells, such as those within the bone marrow [25]. Furthermore, Ma et al. studied patients with ovarian cancer undergoing chemotherapy and found that those who received IVC had significantly lower bone marrow toxicities than those who did not [11]. Their findings suggest that vitamin C may protect hematopoietic progenitor cells from chemotherapy-induced damage, preserve platelet production, and limit thrombocytopenia. Although our results and those of other studies indicate a protective effect of IVC on platelet counts, some studies have reported contrasting results. For example, Wang et al. found no significant difference in thrombocytopenia between vitamin C and control groups in patients with colorectal cancer undergoing chemotherapy [19]. Similarly, Pathak et al. observed no significant impact of IVC on chemotherapy toxicity in patients with lung cancer [20]. These inconsistencies may stem from differences in baseline nutritional status, cancer type, and chemotherapy regimen. In our study, the concurrent use of vitamin C and chemotherapy may have enhanced its protective effects. The mechanism through which vitamin C helps prevent platelet count reduction probably involves reducing oxidative stress and maintaining bone marrow integrity. Chemotherapy-induced oxidative damage is believed to harm megakaryocytes, the precursor cells responsible for platelet production. Vitamin C may neutralize reactive oxygen species (ROS), reduce oxidative stress, preserve megakaryocyte function, and potentially limit excessive platelet depletion [26]. By scavenging ROS, vitamin C mitigates oxidative damage in the bone marrow microenvironment, which is crucial for maintaining the health and function of hematopoietic stem cells and megakaryocytes. Additionally, vitamin C supports endothelial cell function and promotes collagen synthesis, vital for

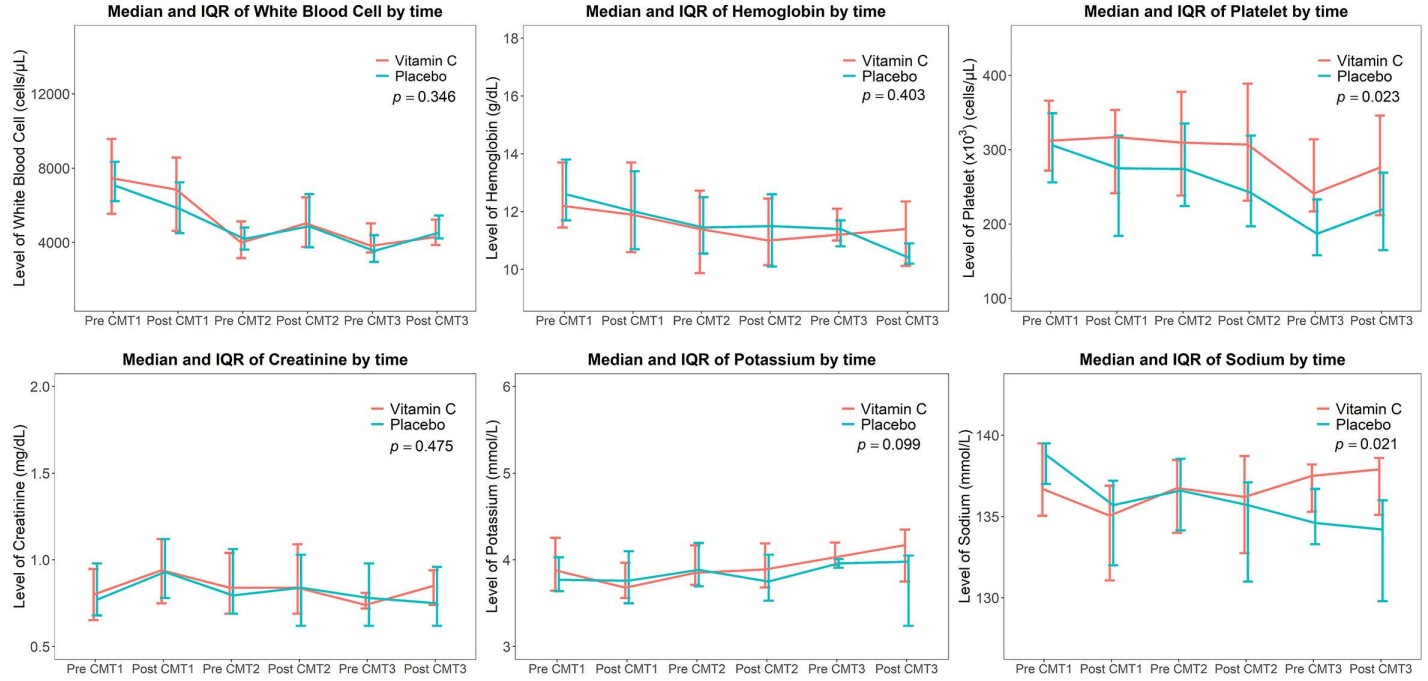

**Fig 2. Changes in hematologic parameters and blood chemistry levels before and after chemotherapy.** Values are presented as median and interquartile ranges (IQRs).

maintaining vascular health and stability. Collagen is crucial for platelet adhesion and aggregation. Therefore, adequate vitamin C levels help maintain the structural integrity of blood vessels, reducing platelet activation and destruction during chemotherapy [27]. Moreover, vitamin C may attenuate the inflammatory responses [28] that contribute to chemotherapy-induced thrombocytopenia, further supporting platelet homeostasis.

In this study, a 2 g dose of IVC administered over 1 hour prior to chemotherapy was selected in accordance with our institution's standardized pre-chemotherapy hydration protocol. IVC is administered as a slow infusion (approximately 33 mg/min) to ensure safety and tolerability [29]. Given the routine 1-hour prehydration period before cisplatin administration, this timeframe permits delivery of up to 2 g without altering standard clinical workflow. This dosing strategy was chosen to evaluate the feasibility and potential supportive effects of IVC within routine clinical practice. This approach is further supported by pharmacokinetic evidence demonstrating that intravenous administration produces markedly higher plasma ascorbate concentrations than oral supplementation and achieves predictable systemic exposure [16]. Additionally, this dosing strategy has been demonstrated to be safe and well tolerated in patients with advanced malignancies, with no reported infusion-related adverse events [25,30]. While significant differences were observed in platelet counts between the IVC and control groups, these values largely remained within normal or near-normal clinical ranges. No overt bleeding episodes were reported during the study period, suggesting limited immediate clinical relevance. However, maintaining more stable platelet counts during chemoradiotherapy may help reduce the risk of bleeding, minimize the need for platelet transfusions, and prevent treatment delays or dose reductions. Such benefits could translate into better tolerance of chemoradiotherapy, improved treatment adherence, fewer unplanned hospital visits, and overall improvements in patients' clinical outcomes. Nevertheless, owing to the exploratory nature of the findings and the limited clinical outcome data, further validation in larger, prospective studies is warranted.

Our study had some limitations. The small sample size may have limited the statistical power to detect subtle or moderate effects of vitamin C on various adverse outcomes, particularly gastrointestinal symptoms and metabolic parameters. Furthermore, the short-term follow-up—limited to three chemotherapy cycles—may not have captured the cumulative effects of treatment. Longer follow-up periods are needed to assess potential long-term benefits or risks. The 2 g dose of vitamin C used may have been insufficient to elicit more pronounced therapeutic effects. Higher doses or more frequent administration might result in greater reductions in other chemotherapy-related side effects. Future studies should address the limitations of the present study by enrolling larger participant groups and extending the follow-up period to evaluate the sustained effects of vitamin C on toxicities. Conducting multicenter randomized controlled trials would enhance the generalizability of the findings and provide greater statistical power. Additionally, investigating alternative dosing regimens of IVC, as well as its potential combined effect with other antioxidants or supportive care agents, may further optimize its therapeutic impact. Collectively, these efforts will help clarify the clinical utility of IVC as an adjunctive therapy in nasopharyngeal carcinoma.

## Conclusions

IVC administration did not significantly reduce gastrointestinal adverse events in patients with nasopharyngeal carcinoma undergoing CCRT. However, it was associated with attenuation of platelet count decline during treatment. As this was a secondary outcome, preservation of platelet levels may help maintain treatment intensity and reduce the risk of bleeding or chemotherapy interruptions. This observation suggests a potential supportive effect of IVC on platelet-related toxicity during CCRT, although confirmatory studies are required.

## Acknowledgments

We are grateful to Ms. Jirawan Jayuphan and Mr. Sittidet Nualnim in the research consultation department for their suggestions and assistance.

## Author contributions

**Conceptualization:** Peesit Leelasawatsuk, Pasawat Supanimitjaroenporn, Nattida Rodsom, Theepat Wongkittithaworn, Manupol Tangthongkum.

**Data curation:** Peesit Leelasawatsuk, Nattida Rodsom, Theepat Wongkittithaworn, Manupol Tangthongkum.

**Formal analysis:** Peesit Leelasawatsuk, Pasawat Supanimitjaroenporn, Nattida Rodsom, Theepat Wongkittithaworn, Manupol Tangthongkum.

**Funding acquisition:** Peesit Leelasawatsuk, Manupol Tangthongkum.

**Methodology:** Peesit Leelasawatsuk, Nattida Rodsom, Manupol Tangthongkum.

**Project administration:** Manupol Tangthongkum.

**Resources:** Peesit Leelasawatsuk.

**Supervision:** Manupol Tangthongkum.

**Validation:** Manupol Tangthongkum.

**Visualization:** Manupol Tangthongkum.

**Writing – original draft:** Peesit Leelasawatsuk, Pasawat Supanimitjaroenporn, Nattida Rodsom, Theepat Wongkittithaworn, Manupol Tangthongkum.

**Writing – review & editing:** Peesit Leelasawatsuk, Manupol Tangthongkum.

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
