## [Decision Letter · Decision Letter 0]

5 Feb 2026

Dear Dr. Tangthongkum,

Thank you for submitting your manuscript to PLOS ONE. After careful consideration, we feel that it has merit but does not fully meet PLOS ONE’s publication criteria as it currently stands. Therefore, we invite you to submit a revised version of the manuscript that addresses the points raised during the review process.

We look forward to receiving your revised manuscript.

Kind regards,

Zhiling Yu

Editor

PLOS One

Journal Requirements:

3. Thank you for submitting your clinical trial to PLOS ONE and for providing the name of the registry and the registration number. The information in the registry entry suggests that your trial was registered after patient recruitment began. PLOS ONE strongly encourages authors to register all trials before recruiting the first participant in a study.

1) your reasons for your delay in registering this study (after enrolment of participants started);

2) confirmation that all related trials are registered by stating: “The authors confirm that all ongoing and related trials for this drug/intervention are registered”.

“This research was financially supported by a grant from the Faculty of Medicine, Prince of Songkla University, Thailand.”

“This research was financially supported by a grant from the Faculty of Medicine, Prince of Songkla University, Thailand.”

We note that one or more of the authors is affiliated with the funding organization, indicating the funder may have had some role in the design, data collection, analysis or preparation of your manuscript for publication; in other words, the funder played an indirect role through the participation of the co-authors. If the funding organization did not play a role in the study design, data collection and analysis, decision to publish, or preparation of the manuscript and only provided financial support in the form of authors' salaries and/or research materials, please do the following:

1. Review your statements relating to the author contributions, and ensure you have specifically and accurately indicated the role(s) that these authors had in your study. These amendments should be made in the online form.

2. Confirm in your cover letter that you agree with the following statement, and we will change the online submission form on your behalf:

“The funder provided support in the form of salaries for authors [insert relevant initials], but did not have any additional role in the study design, data collection and analysis, decision to publish, or preparation of the manuscript. The specific roles of these authors are articulated in the ‘author contributions’ section.

Reviewers' comments:

Reviewer's Responses to Questions

**Comments to the Author**

1. Is the manuscript technically sound, and do the data support the conclusions?

Reviewer #1: Partly

Reviewer #2: No

Reviewer #3: Yes

Reviewer #4: Yes

Reviewer #5: Yes

2. Has the statistical analysis been performed appropriately and rigorously?

Reviewer #1: Yes

Reviewer #2: No

Reviewer #3: Yes

Reviewer #4: Yes

Reviewer #5: No

3. Have the authors made all data underlying the findings in their manuscript fully available?

Reviewer #1: Yes

Reviewer #2: Yes

Reviewer #3: Yes

Reviewer #4: Yes

Reviewer #5: Yes

4. Is the manuscript presented in an intelligible fashion and written in standard English?

Reviewer #1: Yes

Reviewer #2: Yes

Reviewer #3: Yes

Reviewer #4: Yes

Reviewer #5: Yes

Reviewer #1: 1. While vitamin C has shown potential in reducing chemotherapy-induced toxicities in other cancers – Sentence changed to chemotherapy-induced toxicities in some of cancers.

2. Common adverse effects include gastrointestinal symptoms, such as nausea, anorexia, oral mucositis, diarrhea, and dysphagia, hematological toxicities (e.g., neutropenia, anemia, and thrombocytopenia), electrolyte imbalances (e.g., hyponatremia and hypokalemia). – Reference should be include.

3. Recent studies have largely focused on IVC, with doses ranging from 1 to 10 g per session, - How many sessions (its per day or something else? Its needs to be clarify in the manuscript.

4. Sixty-eight patients diagnosed with NPC and scheduled for CCRT were enrolled. – Which stage of NPC patients chosen for this study?

5. Regarding laboratory parameters, reductions in platelet count and serum sodium level were significantly less in the vitamin C group compared with the placebo group. – As of your results in the tables, there’s no difference in vitamin C and placebo groups.

6. However, platelet count reductions were significantly less in the vitamin C group than in the placebo group following chemotherapy – Mention the values in the manuscript.

7. Abbreviations should be include in the manuscript.

8. Quality of figures must be improved.

9. Based on your results there’s not much improvement of an adverse effects from chemotherapy. What is your conclusion about your study?

10. Why are you choosing 2g IVC for your study? As per literature it was 1-10g, please explain these in the manuscript.

Reviewer #2: Introduction

1. Author need to enhance the introduction part

2. Author missed the rational study objection

3. Author provide the FDA toxicological data in the intro part

4. Methodological part need to improve

5. Author need to rewrite result part with statistical comparison

6. Discussion is Very short, author need rewrite mechanistic discussion with result comparison

7. Conclusion not covey the correct way

8. Figure 2 very poor presentation and all the statistical error look like similar, author recheck the statistical error value

Reviewer #3: This study demonstrated that before the CCRT administration, IVC was effective in mitigating the reduction of platelet counts in patients with nasopharyngeal carcinoma (NPC). As the authors mentioned, this study has limitations, including a small patient sample size and short-term follow-up; however, it is very interesting in that it suggests IVC may have clinically meaningful benefits. I suggest that the authors should improve the following comments;

-The abstract contains too much methodological detail. It should be revised to briefly describe the study design and focus on the key findings.

-The manuscript lacks a detailed description of how hematological values were adjusted for individual patients in the graphical representations. Please clarify whether baseline normalization, delta changes, or other statistical adjustments were applied to account for inter-patient variability.

-The introduction should provide more focused literature references regarding vitamin C’s effect on platelet counts in specific cancer types.

-The conclusion should be revised to emphasize the potential clinical benefit of IVC in mitigating platelet count reduction, which would better highlight the significance of this study.

Reviewer #4: This manuscript presents a well-designed randomized, double-blind, placebo-controlled clinical trial addressing an important supportive care question in nasopharyngeal carcinoma. The study is methodologically rigorous, with clear eligibility criteria, appropriate randomization and blinding procedures, and transparent reporting in line with CONSORT guidelines. The authors provide a balanced interpretation of both positive and null findings, particularly highlighting the clinically relevant observation of platelet preservation. Overall, the work adds meaningful evidence to the limited literature on intravenous vitamin C in this setting and is clearly written and well-structured. Hence this manuscript may be recommended.

Reviewer #5: This paper describes what appears to be a well conducted study randomised trial comparing Vitamin-C with Placebo. Nevertheless, there are a few issues as listed below.

1. The primary endpoints are regarded as GI adverse events. So that when justifying the trial size of 22 per group (sensibly increased to 30), although power and test size are specified the anticipated difference between the groups is not. This difference should be specified.

2. Table 1 Patient demographics: As this is a randomised trial then, by definition, any differences observed between characteristics in this table are ‘random’. So, testing statistical differences is wrong and the p-values in the table should be deleted.

3. Table 1: Probably better to describe minimum and maximum age rather than the SD. Also, for the pretreatment values WBC, Haemoglobin, etc.

4 Table 2: For Nausea, the total of Grade 1 and Grade 2 adverse events with Vitamin C sums to greater than 30. So there so there is some double counting which makes the statistical testing invalid. The totals here should then be replaced by (say) the highest grade of adverse event experienced by the patient. The numbers experiencing zero adverse events should also be included to ensure that each sub table sums to 30 in each treatment group.

5. Table 2: In any event, although all may be incorrect because of the double counting, p-values are best quoted to 2 significant figures. So, 0.123 to 0.12 but ‘079’ remains as ‘0.079’, 0.155’ to 0.16’ and so on.

6. Table 2: Also, it would have helped to present, for example, for Oral mucositis (adding the zero group and presuming no double counting) the proportion of each Grade of adverse event in the margins of the sub tables. Here 7/25 = 0.28, 16/23 = 0.70, 3/3 = 0.5 and 4/6 = 0.67.

On a positive note, the authors when presenting results concerning the laboratory parameters refer to a Bayesian logistic regression. This is an unusual addition from a statistical perspective, but if retained needs much more explanation of what the methodology does and how the results are to be interpreted.

.

Reviewer #1: **Yes:** Rajesh SelvarajRajesh SelvarajRajesh SelvarajRajesh Selvaraj

Reviewer #2: No

Reviewer #3: No

Reviewer #4: No

Reviewer #5: No

---

## [Author Response · Author response to Decision Letter 1]

5 Mar 2026

Dear Editor,

We would like to sincerely thank you and the reviewers for the insightful and constructive comments provided on our manuscript. We have carefully addressed all comments from the reviewers and revised the manuscript accordingly. A point-by-point response to each reviewer’s comments is provided as "Response to Reviewers.docx" file, with the corresponding changes made to the manuscript highlighted as "Revised Manuscript with Track Changes.docx" file. We appreciate the opportunity to revise our work and hope that the revised manuscript will now meet the journal's standards.

Sincerely,

Manupol Tangthongkum, MD

(Corresponding author)

---

## [Decision Letter · Decision Letter 1]

24 Mar 2026

Effect of intravenous vitamin C administration on chemotherapy-induced adverse events in patients with nasopharyngeal cancer

PONE-D-25-67982R1

Dear Authors,

We’re pleased to inform you that your manuscript has been judged scientifically suitable for publication and will be formally accepted for publication once it meets all outstanding technical requirements.

Kind regards,

Academic Editor

PLOS One

Additional Editor Comments (optional):

Reviewers' comments:

Reviewer's Responses to Questions

**Comments to the Author**

Reviewer #5: All comments have been addressed

2. Is the manuscript technically sound, and do the data support the conclusions?

Reviewer #5: Yes

3. Has the statistical analysis been performed appropriately and rigorously?

Reviewer #5: Yes

4. Have the authors made all data underlying the findings in their manuscript fully available?

Reviewer #5: Yes

5. Is the manuscript presented in an intelligible fashion and written in standard English?

Reviewer #5: Yes

Reviewer #5: Many thanks to the authors. They have addressed all the points I raised in my earlier review.

MMany thanks to the authors. They have addressed all the points I raised in my earlier review.

Many thanks to the authors. They have addressed all the points I raised in my earlier review.

Many thanks to the authors. They have addressed all the points I raised in my earlier review.

Many thanks to the authors. They have addressed all the points I raised in my earlier review.

.

Reviewer #5: No

---

## [Editor Report · Acceptance letter]

PONE-D-25-67982R1

PLOS One

Dear Dr. Tangthongkum,

I'm pleased to inform you that your manuscript has been deemed suitable for publication in PLOS One. Congratulations! Your manuscript is now being handed over to our production team.

Kind regards,

on behalf of

Dr. Zhiling Yu

Academic Editor

PLOS One